# Is indoor environment a risk factor of building-related symptoms?

**Kayo Tsumura**[1,2], **Hiroko Nakaoka**[2,3], **Norimichi Suzuki**[2], **Kohki Takaguchi**[2], **Yoshitake Nakayama**[2], **Keiichi Shimatani**[2], **Chisato Mori**[2,3]*

**1** Graduate School of Medical and Pharmaceutical Sciences, Chiba University, Chiba, Japan, **2** Center for Preventive Medical Sciences, Chiba University, Chiba, Japan, **3** Department of Bioenvironmental Medicine, Graduate School of Medicine, Chiba University, Chiba, Japan

* cmori@faculty.chiba-u.jp

## Abstract

The indoor environment, particularly indoor air quality (IAQ), is significantly associated with building-related symptoms (BRSs) in humans. In our previous studies, we demonstrated a significant relationship between BRSs and indoor chemical concentrations. In Japan, the Ministry of Health, Labor, and Welfare (MHLW) guideline recommends an air quality target of 13 volatile organic compounds (VOCs) and a provisional target of 400 µg/m$^3$ for total VOCs (TVOC). The objective of this study was to determine the relationship between TVOC levels and the risk of BRSs using the Japanese provisional target TVOC level of 400 µg/m$^3$. The relationship between odor intensity and BRSs while the TVOC levels were under 400 µg/m$^3$ was also examined. The study was conducted in a laboratory house (LH) on the campus of Chiba University from 2017–2019. The study included 149 participants who spent 60 minutes in the LH. The participants were asked to evaluate the IAQ of the LH. A significant relationship between the risk of BRSs and the provisional target TVOC level was observed (odds ratio: 2.94, 95% confidence interval: 1.18–7.35). Furthermore, a significant relationship between odor intensity and risk of BRSs in spaces with TVOC levels less than 400 µg/m$^3$ was detected (odds ratio: 6.06, 95% confidence interval: 1.21–30.44). In conclusion, the risk of BRSs is significantly lower in spaces with low TVOC levels and low odor intensity. Reducing the concentration of airborne chemicals and odor intensity may improve IAQ and prevent BRSs.

## Introduction

People spend about 90% of their day indoors [1], and indoor air quality (IAQ) is closely related to human health. The United States Environmental Protection Agency defines IAQ as the quality of the air inside and around buildings and structures, especially related to the health and comfort of building occupant [2]. IAQ is mainly assessed based on the concentration of pollutants and chemicals in the air such as ambient fine particulate matter, carbon dioxide, and volatile organic compounds (VOCs). The World Health Organization (WHO) provides guidelines for VOCs commonly present in indoor air [3]. In Japan, the Ministry of Health,

**Data Availability Statement:** Our data are unsuitable for public deposition due to ethical restrictions and the legal framework of Japan. It is prohibited by the Act on the Protection of Personal

Information (Act No. 57 of 30 May 2003, amendment on 9 September 2015) to publicly deposit data containing personal information. Ethical Guidelines for Medical and Health Research Involving Human Subjects enforced by the Japan Ministry of Education, Culture, Sports, Science and Technology and the Ministry of Health, Labour and Welfare also restrict the open sharing of epidemiologic data. All inquiries about access to data should be sent to: suzu-nori@chiba-u.jp or ho-cpms@office.chiba-u.jp.

**Funding:** The study was supported by a grant from the Sekisui House Ltd. and Japan Society for the Promotion of Science (JSPS) Grants-in-Aid for Scientific Research (C) Grant Number (19K12455, 18K13885). This work was supported by JST OPERA Program, Japan (Grant Number JPMJOP1831). The sponsor had no control over the interpretation, writing, or publication of this work. The funders had no role in study design, data collection and analysis, decision to publish, or preparation of the manuscript.

**Competing interests:** The author of this article, Kayo Tsumura, and the co-authors Yoshitake Nakayama, Koki Takaguchi, and Keiichi Shimatani are affiliated with the Endowed Chair of Sekisui House, Ltd. at the Centre for Preventive Medicine, Chiba University. The other authors declare that they have no known competing financial interests or personal relationships that would affect the research reported in this paper. This does not alter our adherence to PLOS ONE policies on sharing data and materials.

Labour and Welfare (MHLW) also set guideline concentration values for 13 VOCs and a provisional target value of 400 μg/m$^3$ for total VOC (TVOC) level as measures against the effects of VOCs on the health of humans [4,5]. In our previous study, we found a significant relationship between indoor chemical concentrations and the occurrence of building-related symptoms (BRSs) [6]. The study findings suggest that individuals with a history of allergy and high sensitivity to chemicals are more likely to have BRSs. These findings indicate that human reactions to indoor air pollutants depend on the levels of airborne chemicals and on several other factors, including pre-existing medical conditions and personal sensitivity.

In previous studies, BRSs were found to be associated with odor. Odor perception is an early predictor of BRSs, and it is attributed to one or more VOCs [7]. Odor perception is affected by many different types of odor.

The type and intensity of odors are primarily evaluated based on individual olfactory responses. Measurements based on human olfactory response is the most common method of odor concentration evaluation [8], but the intensity and sensation of odors are different for each individual. People perceive odors due to one or more VOCs, and these substances are present in various concentrations in the air. Odors help with the evaluation of IAQ and avoidance of exposure to harmful substances. Some VOCs cause eye and airway or upper respiratory tract irritation and symptoms [9]. The IAQ and odor of real spaces are due to multiple chemicals. It is difficult to identify factors associated with health risk because multiple chemicals and odor substances coexist in the indoor air environment [10]. Our previous study showed that reducing the sum of VOC (ΣVOC) levels and odor intensity in indoor air promotes human relaxation [11]. The aims of this study were (1) to examine whether the Japanese provisional target TVOC level of 400 μg/m$^3$ was effective in reducing the risks of BRSs and (2) to investigate the association between odors and human health to determine if odor can be considered a new indicator of IAQ and its human health effects.

## Materials and methods

### Study site

This study was conducted at a laboratory house (LH) built as a study site in November 2017 on the Kashiwa-no-ha campus of Chiba University. The LH has one living room and two bedrooms and is furnished. The structure of the LH is a typical Japanese wooden house constructed from timber. The LH was constructed mostly with F☆☆☆☆ (F-four stars) labeling. The F☆☆☆☆ rating is a classification of building materials in Japan based on the formaldehyde emission rate, which is defined by Japanese Industrial Standards (JIS), Japanese Agricultural Standards (JAS), and the Ministry of Land, Infrastructure, Transport, and Tourism of Japan. F-four-stars building materials, including boards, adhesives, insulation, and paints, can be used without limitations on the area of use. The LH was equipped with a mechanical ventilation system designed to ventilate 0.5 times per hour. The LH was also equipped with air conditioners that control the temperature and humidity, but no air purifiers were used. The rooms of the LH were not decorated with any indoor plants.

### Participants and study design

This study was conducted between May and October of the years 2017 to 2019. Participants were recruited to evaluate the IAQ of the LH based on their sensory perception for a period of 60 minutes. All the participants were ordinary people such as students, workers, and housewives, and they were not previously trained for the study. They were recruited in pairs or were asked to invite a friend or acquaintance to pair with for the study. On the day of the test, the aims and the test procedure were explained to the participants, and written informed consent

was obtained from all the participants. The body temperature, blood pressure, and pulse of the participants were then measured as a health check.

The test was conducted. During the 60 minutes of the experiment, two participants first stayed together in the living room for 30 minutes, and then proceeded to each spend 30 minutes separately in the bedrooms. The participants received guidance regarding the test and how to complete the questionnaires via audio announcements in each bedroom. The participants completed the online questionnaire using the Questant software (Macromill Inc., Tokyo, Japan) installed on a tablet. In each room, they first spent five minutes in the indoor environment to acclimatize to the indoor air and odor. After that, they indicated their impressions of the room air and odor by responding to the corresponding questions in the questionnaire. The participants spent 20 minutes freely in the room after answering questions regarding their characteristics and medical history in the questionnaire. Thereafter just before leaving the room, they completed a questionnaire on their physical condition and the occurrence of BRSs.

This study protocol was approved by the Research Ethics Committee of the Graduate School of Chiba University (No.2737). The study was conducted in accordance with the principles of the World Medical Association Declaration of Helsinki.

## Questionnaire

During the experiment, participants were asked to complete several questionnaires. The questionnaire includes questions on odors perceived, personal characteristics, indoor environment of the building where the participants spend most of the day, indoor environments that may impact health, awareness of air quality, and BRSs. Further, the Quick Environmental Exposure and Sensitivity Inventory (QEESI) was used to determine the sensitivity of participants to chemicals [12,13]. QEESI was developed by Miller and Prihoda in the USA to determine the causal relationship between chemicals and personal sensitivity. Then, using the cut-off value for Japanese individuals reported by Hojo et al. [14], we determined the sensitivity or non-sensitivity of participants to chemicals. A participant was considered highly susceptible (i.e., with high QEESI score) if at least two of the following three criteria are observed: symptoms $\geq 20$ points, chemical intolerance $\geq 40$ points, and degree of daily life disturbance $\geq 10$ points. Finally, using the questionnaire referred to in the Consultation Manual for Trace Chemicals in the Indoor Environment issued by the Japanese MHLW, participants were asked about the symptoms they experienced during the test.

Participants answered questions regarding the site of symptoms (e.g., mucous membrane of the eyes, nose, throat, or airways, or the skin, muscles, joints, or digestive tract). We diagnosed them using a questionnaire developed based on the "Consultation Manual for Trace Chemicals in the Indoor Environments"issued by the Japanese MHLW. Symptom severity was rated on a four-point scale as follows: 1. no symptoms;2. mild symptoms;3. moderate symptoms; and 4. severe symptoms. Participants were asked to rate the severity of their symptoms. Participants were considered to have BRSs ("yes") if at least eight questions were rated 2, 3, or 4 by participants: otherwise, participants were considered to have no BRSs ("no").

Furthermore, the participants were asked to subjectively score their health status on the day of the evaluation using a scale of 1–4: 1. very good; 2.good; 3.not good; or 4.bad. We defined 1 and 2 as "good" and 3 and 4 as "bad". We classified scores of 1 and 2 as "good" and scores of 3 and 4 as "bad." Regarding allergies, participants were asked to indicate if they had been diagnosed by a physician with any of the following conditions: asthma, atopic dermatitis, allergic rhinitis, allergic conjunctivitis, food allergy, or urticaria. A medical history of allergy was noted if a participant selected at least one condition. Participants were also asked to rate their smoking status on a scale of 1 to 4 as follows: 1. never smoked;2. stopped smoking more than five

years ago;3. stopped smoking within the last four years; or 4. still smoking. Participants who indicated 4 were designated as current smokers.

## Collection of indoor air samples and analysis

The indoor air of the LH was sampled on the morning of each test. During air sampling, room temperature and humidity were measured using a thermo-hygrometer data logger (SK-L200THIiα; Sato Keiryoki Mfg. Co. Ltd., Tokyo, Japan). A mechanical ventilation system was installed in the LH and set to ventilate 0.5 times per hour. Standard air sampling methods issued by the Japanese MHLW (MHLW 2000) were used in this study. Before air sampling, the windows and doors of the LH were kept open for 30 minutes for ventilation. Indoor air samples were collected for 30 minutes at height of 1.2 m in the middle of the living room using active sampling pumps (Shibata MP-Σ30N and MP-Σ100HN; Shibata Scientific Technology Ltd., Saitama, Japan). VOCs were thermally desorbed from the sampled tube and analyzed using a thermal desorber (TurboMatrix650; PerkinElmer, USA) coupled with a gas chromatograph (7890B GC system; Agilent, USA)/mass spectrometer (5977A MSD GC/MS; Agilent, USA). Carbonyl compounds were eluted with acetonitrile from the gas tubes and analyzed using a high-performance liquid chromatograph (Prominence HPLC system; Shimadzu, Japan). Details of the chemical analysis were presented in our previous study [15]. The target compounds in this study were 51 VOCs and 2 carbonyl compounds. They were identified and analyzed. The limit of quantification for each chemical was 1.0 μg/m$^3$. ΣVOC level in this study was defined as the sum of the concentrations of identified VOCs and carbonyl compounds.

## Calculation of odor activity value (OAV) and the sum of OAVs (SOAV)

A method of calculating OAV [16] was used to quantify the odor of each VOC in indoor air. OAV was determined as the ratio of the measured concentration of a particular odor chemical in the air to its odor threshold concentration. In other words, OAV was calculated as the concentration of a VOC in indoor air divided by its odor threshold concentration. The odor threshold concentrations used in this study were published by the Japanese Ministry of Environment [17]. The concentration of each VOC in the air was converted to OAV using equation 1 below, and SOAV was calculated using equation 2.

1. OAV = concentration of each chemical (μg/m$^3$)/odor threshold value of each chemical (μg/m$^3$)

2. SOAV = sum of OAVs.

## Statistical analysis

All analyses were performed using the statistical software package SPSS version 25.0 for Windows (SPSS Inc., Chicago, IL, USA). We first categorized spaces with ΣVOC levels of $< 400$ μg/m$^3$ as Case 1 and spaces with ΣVOC level of $> 400$ μg/m$^3$ as Case 2 to verify the provisional target TVOC level of 400 μg/m$^3$ set by the Japanese MHLW. Then, the relationship between ΣVOC level and the occurrence of BRSs was evaluated using logistic regression analysis after adjusting for environmental factors (e.g., temperature, relative humidity) and characteristics of participants in Case 1 and Case 2. The relationship between odor (SAOV) and the occurrence of BRSs was also evaluated in Case 1. Covariates were selected based on our previous study. In addition, we determined final covariates from variables using Mann–Whitney U test. In the analysis, odds ratios (ORs) and 95% confidence intervals (CIs) were calculated, and $p < 0.05$ was considered significant.

## Results

### Characteristics of the study population and symptoms

Table 1 shows the characteristics of the study participants. One hundred and twenty-five individuals (Females: 51 participants [40.8%], Males: 74 participants [59.2%]) participated, and their data are shown after grouping based on ΣVOC level into Case 1 or Case 2. Eighty-nine participants (71.2%) were in their 20s, while 19 participants (15.2%) were in their 30s (Case 1: 41 participants [87.2%] in their 20s, five [10.6%] in their 30s; Case 2: 48 participants [61.5%] in their 20s, 14 [17.9%] in their 30s). Based on the QEESI questionnaire, a total of 44 participants (35.2%) (Case 1: 16 [34.0%], Case 2: 28 [35.9%]) were sensitive to chemicals. On the day of the test, 114 participants (91.2%) (Case 1: 44 [93.6%], Case 2: 70 [89.7%]) stated that they were in a

**Table 1. Characteristics of participants and indoor environment conditions.**

| Personal factors | | Case 1 | | Case 2 | | P value | |
|---|---|---|---|---|---|---|---|
| | | ΣVOC level ≤ 400 μg/m³ | n = 47 | ΣVOC level > 400 μg/m³ | n = 78 | | |
| | | n | % | n | % | | |
| Sex | | | | | | | |
| | Male | 29 | 61.7 | 45 | 57.7 | 0.660 | |
| | Female | 18 | 38.3 | 33 | 42.3 | | |
| Age (years) | | | | | | | |
| | 20–29 | 41 | 87.2 | 48 | 61.5 | **0.001** | * |
| | 30–39 | 5 | 10.6 | 14 | 17.9 | | |
| | 40–49 | | | 9 | 11.5 | | |
| | ≥50 | 1 | 2.1 | 7 | 9.0 | | |
| Sensitivity to chemicals (QEESI) | | | | | | | |
| | Low | 31 | 66.0 | 50 | 64.1 | 0.834 | |
| | High | 16 | 34.0 | 28 | 35.9 | | |
| Physical condition | | | | | | | |
| | Good | 44 | 93.6 | 70 | 89.7 | 0.461 | |
| | Not Good | 3 | 6.4 | 8 | 10.3 | | |
| Medical history of allergy | | | | | | | |
| | No | 17 | 36.2 | 35 | 44.9 | 0.341 | |
| | Yes | 30 | 63.8 | 43 | 55.1 | | |
| Current smoking | | | | | | | |
| | No | 36 | 76.6 | 66 | 84.6 | 0.264 | |
| | Yes | 11 | 23.4 | 12 | 15.4 | | |
| Occurrence of BRSs | | | | | | | |
| | No | 33 | 70.2 | 36 | 46.2 | **0.009** | * |
| | Yes | 14 | 29.8 | 42 | 53.8 | | |
| **Environmental factors** | | Case 1 | | Case 2 | | P value | |
| | | Mean | SD | Mean | SD | | |
| Temperature | °C | 23.02 | 0.82 | 23.77 | 1.26 | **<0.001** | * |
| Humidity | % | 55.95 | 5.68 | 56.92 | 12.42 | **0.002** | * |
| ΣVOC level | μg/m³ | 278 | 52 | 2980 | 1982 | **<0.001** | * |
| SOAV | | 7.0 | 1.7 | 30.2 | 20.8 | **<0.001** | * |

*Statistically significant (i.e., p < 0.05).

VOC: Volatile organic compound, QEESI: Quick Environmental Exposure and Sensitivity Inventory, BRSs: Building-related symptoms, SD: Standard deviation, SOAV: Sum of odor activity values.

"good" state of health. Further, 73 participants (58.4%) (Case 1: 30 [63.8%], Case 2: 43 [55.1%]) had a history of allergy. Fifty-six participants (44.8%) (Case 1: 14 [29.8%], Case 2: 42 [53.8%]) had BRSs; Thus, the number of participants with BRSs was greater in Case 2 than in Case 1. Except for age, no participant characteristic was significantly different between Case 1 and Case 2.

### Environmental measurements

The IAQ data (for Case 1 and Case 2, including temperature and humidity) are shown in Table 1. The environmental data of Case 1 and Case 2 were compared using Mann–Whitney U test. The results showed significant differences in temperature, humidity, ΣVOC level, and SOAV between Cases 1 and 2. S1 and S2 Tables (S1 and S2 Tables) show the substances identified and analyzed. In Case 1, 27 substances were identified, and 16 of them had a frequency of 100%. In Case 2, 47 substances were identified, and 12 of them had a frequency of 100%. The OAVs of nine substances in Case 1 and 18 substances in Case 2 were calculated.

### Association between the occurrence of BRSs and ΣVOC level

In this study, 56 participants (44.8%) (Case 1: 14 [29.8%], Case 2: 42 [53.8%]) had BRSs, and there was a significant difference in the occurrence of BRSs between Case 1 and Case 2 ($p < 0.01$; Table 1). Table 2 shows the results of the multivariate logistic regression analysis. In the analysis, covariates were determined using Mann–Whitney U test. Since there were significant differences in temperature and humidity between Cases 1 and 2, we divided the variables into two categories based on their median values (i.e., 23.5°C and 58.8%, respectively) and included them as covariates. The results show that the occurrence of BRSs is significantly associated with ΣVOC level of >400 μg/m$^3$ (OR: 2.94, 95% CI: 1.18–7.35).

### Association between odor (SAOV) and occurrence of BRSs

Fourteen participants in Case 1 (29.8%) had BRSs. These participants had BRSs even at ΣVOC level ≤ 400 μg/m$^3$. To investigate the relationship between odor (SAOV) and the occurrence

**Table 2. Logistic regression analysis of personal and environmental factors and occurrence of BRSs.**

| | | P value | | OR | 95% CI | | |
|---|---|---|---|---|---|---|---|
| Temperature | | | | | | | |
| | ≤23.5 | | | Ref | | | |
| | ≥23.5 | 0.246 | | 1.60 | 0.72 | — | 3.53 |
| Humidity | | | | | | | |
| | ≤58.75 | | | Reference | | | |
| | ≥58.75 | 0.135 | | 0.53 | 0.23 | — | 1.22 |
| Age (years) | | | | | | | |
| | 20–29 | 0.819 | | Ref | | | |
| | 30–39 | 0.533 | | 0.71 | 0.25 | — | 2.10 |
| | 40–49 | 0.536 | | 1.61 | 0.38 | — | 7.28 |
| | ≥50 | 0.930 | | 1.07 | 0.21 | — | 5.09 |
| ΣVOC level | | | | | | | |
| Case 1 | ≤400 | | | Ref | | | |
| Case 2 | >400 | **0.021** | * | **2.94** | **1.18** | — | **7.35** |

*p < 0.05. VOC: Volatile organic compound, OR: Odds ratio, CI: Confidence interval, BRSs: Building-related symptoms.

**Table 3. Logistic regression analysis of SOAV and occurrence of BRSs.**

|  |  | P value |  | OR | 95% CI |  |  |
|---|---|---|---|---|---|---|---|
| SOAV |  |  |  |  |  |  |  |
| Low | ≤6.9 |  |  | Ref |  |  |  |
| High | >6.9 | 0.029 | * | 6.06 | 1.21 | − | 30.44 |

Other covariates (such as temperature, humidity, and age) were adjusted.

*p < 0.05.

SOAV: Sum of odor activity values, BRSs: Building-related symptoms, OR: Odds ratio, CI: Confidence interval.

of BRSs in Case 1, SOAV in Case 1 was divided into two groups based on the median SOAV of 6.9 and added as a covariate to the multivariate logistic regression analysis. The results of the analysis are shown in Table 3. The results show that the occurrence of BRSs is significantly associated with high SOAV (OR: 6.06, 95% CI: 1.21–30.44).

## Discussion

In this study, we examine the effectiveness of the Japanese provisional target TVOC level ($400 \ \mu g/m^3$) in reducing the risk of BRSs. We also evaluated the impact of odor intensity on human health at low TVOC levels. The number of participants who reported BRSs during their stay at the LH was 56 (44.8%). The reported prevalence of BRSs (or sick building syndrome) in Asia-Pacific countries is 41.0% in Hong Kong [18], 26.2% in Taiwan [19], and 24.9% in Japan [20]. In our 2017 study [21], which involved a web survey of 1500 Japanese participants, we found that the prevalence of BRSs in Japan was 18.8%. The higher prevalence of BRSs reported in this study than in the above studies may be due to the inclusion of individuals with mild BRS symptoms in the eyes, nose, throat, airways, skin, muscles, joints, and digestive organs [6]. In this study, we found that ΣVOC level of <400 $\mu g/m^3$ is significantly associated with a low prevalence of BRSs. Although TVOC has been used as an indicator of IAQ for over 30 years [22,23], the use of TVOC is still controversial. Shrubsole et al. concluded that individual VOC guidelines are more appropriate than the TVOC guideline**s** [24]. According to Salthammer, T., TVOC cannot be used for health- and odor-related issues [25]. Some investigators argue that health-related effects of air quality should be assessed using specific VOCs rather than TVOC, as TVOC does not reveal the toxicity of individual VOCs to humans [26,27]. However, the concentrations of single VOCs may not be valid measures of health risks [28]. Several studies demonstrated a positive correlation between TVOC or ΣVOCs and symptom severity and prevalence in humans [29,30]. TVOC measurement and analysis methods for research are improving [31]. The association between ΣVOC levels and the low prevalence of BRSs, shown in this study supports the use of TVOC or ΣVOCs as health-related indicators and can be viewed as practical evidence to support the provisional target TVOC level set by the Japanese MHLW. In addition, this study revealed that odor may influence the occurrence of BRSs in spaces with VOC concentration of <400 $\mu g/m^3$. In this study, we used OAV to assess odor and to quantify the odor of each VOC in indoor air. OAV is used worldwide to evaluate odor pollution from various sources, such as industrial areas, piggeries, and landfill sites. It is also used to evaluate odor intensity levels and the impact of each odor [32,33]. There is a strong correlation between SOAV and ΣVOC concentration because OAV is calculated from VOC concentrations. However, the contribution of airborne chemicals to odor in indoor air may differ from the percentage of airborne chemical concentration because of differences in odor characteristics and odor threshold concentration of each substance. Some studies showed that the compositional percentage of airborne chemical concentrations may be different from the

percentage of chemicals contributing to odor [34]. It follows from this report that airborne chemicals with high concentrations do not always dominate the odor profile. As shown in S1 Table, 16 VOCs were detected and identified in spaces with VOC concentrations of <400 μg/ $m^3$. Regarding OAV, acetaldehyde and nonanal were identified as the main odorous substances. The guideline value for acetaldehyde is set at 0.03 ppm (48 μg/$m^3$) by the Japanese MHLW and at 0.17 ppm (300 μg/$m^3$) by the WHO. In contrast, no guideline values have been provided for nonanal. Nonanal is a fragrance component of flowers and fruits (e.g., roses and citrus fruits) and is used in food and cosmetic preparations. It is a type of low-toxicity aldehyde with irritating effects on the eyes, skin, and respiratory organs [35]. In this study, the maximum concentration of acetaldehyde detected in spaces with VOC concentrations of <400 μg/ $m^3$ was 15.10 μg/$m^3$. This means that in almost all cases, the concentration of acetaldehyde was less than the guideline value set by the Japanese MHLW. Nonanal has a lower VOC concentration and OAV than acetaldehyde. Nevertheless, acetaldehyde and nonanal may have contributed significantly to OAV and influenced the occurrence of BRSs. However, decreasing VOC concentrations may not necessarily improve IAQ.

Odor, especially unpleasant odor, is considered a warning sign or indicator of risk to human health [36]. Kjaergaard et al. and Wang et al. reported that stimulation irritation is associated with humidity and dryness [37,38]; in addition, it is associated with odor perception. A study of houses in a large Chinese city showed that odor complaints are a sign of air pollution [39], and it was reported in another study that odor perception is an early predictor of BRSs [7]. These studies show that olfactory sensation acts as a warning sign of decreased IAQ, and they suggest an association between olfactory sensation and BRSs. However, olfactory perception may be affected by adaptation to and preference for odor. We tend to perceive air quality as "clean" when we like the smells (i.e., the air smells nice). When we feel comfortable, the scent tends to reduce health risks [40,41]. However, even at low concentrations, the use of fragrances and deodorants releases chemicals into the air. These fragrances and deodorants may mask unpleasant odors, like odorant chemicals, and they could have unintended effects and risks to human health in the indoor environment. Moreover, olfactory perception is easily fatigued, and when it adapts to an odor, the odor becomes difficult to perceive. The easy fatigability of olfactory perception refers to the ease of adaptation of olfaction to a previously perceived odor. In previous studies examining odor intensity and olfactory adaptation and symptom levels, a negative linear correlation between odor acceptability and symptom levels, and a good positive linear correlation between odor intensity and symptom levels [42]. OAV compensates for these weaknesses in olfactory perception and can be used to evaluate odor intensity. Thus, OAV may be considered a new objective method of IAQ evaluation.

Regarding the evaluation tests of indoor environments including odor perception conducted in the LH, this study has several strengths. First, the study was conducted in a full-scale living environment. This was a significant advantage in odor evaluation experiments. Second, participants focused on IAQ and the occurrence of BRSs during their short stay, which may have allowed them to clearly recognize the development or change of symptoms in the environment they were exposed to for this study. This study also has some limitations. First, VOC levels in the air change constantly, and they change with seasons of the year and with daily lifestyle. Second, the results of this study may not be generalized.

The study results are based on the chemical composition of the air quality and the odor composition in the LH. Most study participants were in their 20s or 30s and included college students. The reason for this choice of participants is that the test had to be conducted during the day on a weekday. Furthermore, given that the occurrence of BRSs is based on the responses of the study participants, we cannot disregard the influence of individual differences on the evaluation of symptoms and odor. Moreover, we did not investigate the effects of long-

term exposure to VOCs in the indoor environment on the study participants. Thus, we do not know the effects of VOCs, even at low concentrations, on exposed individuals for a long time. Recently, as a result of the coronavirus disease 2019 (COVID-19) pandemic, increasingly more time has been spent at home. In Japan, the time people spend indoors increased by 13.8% from before the COVID-19 pandemic [43]. There is growing concern regarding indoor air quality (IAQ) worldwide, with some studies suggesting that IAQ worsens with increased time at home due to the COVID-19 pandemic [44]. We hope that measures will be taken to prevent the IAQ from getting worse and that more studies on this topic will be conducted in the future.

## Conclusions

To the best of our knowledge, this study is the first study to investigate the association between the occurrence of BRSs and differences in the levels of TVOC and the odor intensity under the same environmental conditions. The risk of BRSs is significantly lower in spaces with $\Sigma$VOC concentration of $<400$ μg/m$^3$ than in spaces with $\Sigma$VOC concentration of $>400$ μg/m$^3$. Furthermore, even in spaces with $\Sigma$VOC concentration of $<400$ μg/m$^3$, high odor intensity increases the risk of BRSs, highlighting the need to reduce the concentration of chemicals in the air and to reduce odor intensity. The relationship between TVOC or $\Sigma$VOC concentration, and the occurrence of BRSs is complex, and it is difficult to estimate the combined risk of BRSs. In addition, the mechanism of how the odor triggers symptoms is still unclear. However, our study findings show that OAV can be considered a new method of IAQ evaluation and a new indicator for the prevention of BRSs. Studies focused on odors in indoor air, in addition to those of chemical concentrations, could help improve IAQ and prevent BRS.

## Supporting information

**S1 Table. VOCs and OAV (Case 1).** This table is the substances identified and analyzed in Case1.
(DOCX)

**S2 Table. VOCs and OAV (Case 2).** This table is the substances identified and analyzed in Case2.
(DOCX)

## Author Contributions

**Conceptualization:** Kayo Tsumura, Hiroko Nakaoka, Norimichi Suzuki, Yoshitake Nakayama.

**Data curation:** Kayo Tsumura, Norimichi Suzuki, Kohki Takaguchi.

**Formal analysis:** Kayo Tsumura, Hiroko Nakaoka, Norimichi Suzuki, Keiichi Shimatani.

**Funding acquisition:** Chisato Mori.

**Investigation:** Kayo Tsumura, Norimichi Suzuki, Yoshitake Nakayama.

**Methodology:** Kayo Tsumura, Hiroko Nakaoka, Norimichi Suzuki, Kohki Takaguchi, Keiichi Shimatani.

**Project administration:** Hiroko Nakaoka, Norimichi Suzuki, Yoshitake Nakayama, Chisato Mori.

**Visualization:** Kayo Tsumura.

**Writing – original draft:** Kayo Tsumura.

**Writing – review & editing:** Hiroko Nakaoka, Chisato Mori.

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
