## [Decision Letter · Decision Letter 0]

17 Oct 2022

PONE-D-22-23122Is odor a risk factor of building-related symptoms?PLOS ONE

Dear Dr. Tsumura,

Thank you for submitting your manuscript to PLOS ONE. After careful consideration, we feel that it has merit but does not fully meet PLOS ONE’s publication criteria as it currently stands. Therefore, we invite you to submit a revised version of the manuscript that addresses the points raised during the review process.

We look forward to receiving your revised manuscript.

Kind regards,

Rajeev Singh

Academic Editor

PLOS ONE

Journal Requirements:

"The study was supported by a grant from the Sekisui House Ltd. and Japan Society for the Promotion of Science (JSPS) Grants-in-Aid for Scientific Research (C) Grant Number (19K12455, 18K13885), by JST OPERA Program Grant Number JPMJOP1831, and by JSPS KAKENHI Grant Numbers JP21H01487. The sponsor had no control over the interpretation, writing, or publication of this work."

"The study was supported by a grant from the Sekisui House Ltd. and Japan Society for the Promotion of Science (JSPS) Grants-in-Aid for Scientific Research (C) Grant Number (19K12455, 18K13885), by JST OPERA Program Grant Number JPMJOP1831, and by JSPS KAKENHI Grant Numbers JP21H01487. The sponsor had no control over the interpretation, writing, or publication of this work."

"The authors of this article K.T. (Kayo Tsumura), and co-authors of this Y.N. and K.T. (Kohki Takaguchi), and K.S. are affiliated with the Endowed Chair of Sekisui House, Ltd. at the Centre for Preventive Medicine, Chiba University. The other authors declare that they have no known competing financial interests or personal relationships that would affect the research reported in this paper."

Additional Editor Comments:

Dear Dr. Kayo Tsumura,

Thank you so much for submitting your research to PLOS ONE. In my opinion your manuscript has logical data and some innovative findings. Please find the comments from reviewers and incorporate the required changes in order to get accepted. I look forward to receive your revised manuscript soon.

Thanking You

Dr. Rajeev Singh

Editor PLoS One

Reviewers' comments:

Reviewer's Responses to Questions

**Comments to the Author**

1. Is the manuscript technically sound, and do the data support the conclusions?

Reviewer #1: Partly

Reviewer #2: Yes

2. Has the statistical analysis been performed appropriately and rigorously? 

Reviewer #1: Yes

Reviewer #2: Yes

3. Have the authors made all data underlying the findings in their manuscript fully available?

Reviewer #1: Yes

Reviewer #2: Yes

4. Is the manuscript presented in an intelligible fashion and written in standard English?

Reviewer #1: Yes

Reviewer #2: Yes

5. Review Comments to the Author

Reviewer #1: Title should be change, odor in title replace with indoor environment.

COVID-19 portion remove from manuscript as this study conducted 2017-2019. Focus only VOCs and biological agents which you have studied and data.

Please explain why authors use only 2 case study, as in both cases sample size also different (in case 2- sample size is double), for comparative conclusion, must have study with 3 or 5 cases

Line 83-85: give the detail information of study site area, have any natural air purifier (indoor plants) artificial air purifier inside the rooms and other building construction material details (if available), It also effect the indoor environment quality.

Line 89-93: specify age group, gender in text also

Line 132-135: this data only questionnaires base please justify with diagnostic based results.

Line 199-201: In which parameters used for concluding results as health is good, explain clearly.

Line 201: history of allergy- which types of reports you have for individuals motioned.

In table shown the data for personal factors, one of them physical condition most of them health and not any major health issues reported, until 50 % of them are have medical history of allergy. In pervious mentioned all individuals participate in this study are healthy explain it.

Discussion section updated with recent and more related paper; this portion need improvement.

Rewrite abstract and conclusion section.

Reviewer #2: The work described in the manuscript is based on questionnaire, air sampling and odor activity testing. Data are methodically and accurately analyzed. The conclusions are made basically reflecting to the results formulated.

6. PLOS authors have the option to publish the peer review history of their article (what does this mean?). If published, this will include your full peer review and any attached files.

Reviewer #1: No

Reviewer #2: No

---

## [Author Response · Author response to Decision Letter 0]

13 Dec 2022

Reviewer ＃1

Thank you for your thorough consideration of our manuscript and for providing constructive feedback. The quality of our manuscript has certainly improved as a result of your comments.

We have highlighted all of our changes in the revised manuscript in red. A point-by-point response to the reviewers’ comments is given below.

Comment: The authors greatly appreciate your consideration of our manuscript and helpful comments. We divided your advice into a few points and have addressed each of these suggestions in the responses provided below.

Comment 1: Title should be change, odor in title replace with indoor environment.

Our reply:

Thank you for your advice. We changed the title to “Is indoor environment a risk factor for building-related symptoms?”.

Comment 2: COVID-19 portion remove from manuscript as this study conducted 2017-2019. Focus only VOCs and biological agents which you have studied and data.

Our reply:

Thank you for the comment. We removed the section about COVID-19 from the Introduction and focused only on VOCs and other factors.

Comment 3: Please explain why authors use only 2 case study, as in both cases sample size also different (in case 2- sample size is double), for comparative conclusion, must have study with 3 or 5 cases.

Our reply:

Thank you for the comment. In this study, we tried to examine the provisional target value of 400 µg/m3 set by the Japanese Ministry of Health, Labor, and Welfare. Thus, we divided the scenes into two: the scene with TVOC > 400 µg/ m3 is referred to as Case 1, and the second scene with TVOC < 400 µg/ m3 is referred to as Case 2. The number of participants was not the same as they were divided by TVOC concentration values. We revised the aim of this study in the Introduction.

Comment 4: Line 83-85: give the detail information of study site area, have any natural air purifier (indoor plants) artificial air purifier inside the rooms and other building construction material details (if available), It also effect the indoor environment quality.

Our reply:

Thank you for the comment. As you said, some devices and things affect indoor air quality. We added details of the study site in the text as follows:

“The structure of the LH is a typical Japanese wooden house constructed from timber. The LH was constructed mostly with F☆☆☆☆ (F-four stars) labeling. The F☆☆☆☆ rating is a classification of building materials in Japan based on the formaldehyde emission rate, which is defined by Japanese Industrial Standards (JIS), Japanese Agricultural Standards (JAS), and the Ministry of Land, Infrastructure, Transport, and Tourism. Four-star building materials, including boards, adhesives, insulation, and paints, can be used without limitations on the area of use.

The LH was equipped with a mechanical ventilation system designed to ventilate 0.5 times per hour. The LH was also equipped with air conditioners that control the temperature and humidity, but no air purifiers were used. The rooms of the LH were not decorated with any indoor plants.”

Comment 5: Line 89-93: specify age group, gender in text also

Our reply:

Thank you for the comment. We have specified age groups and gender in the text.

Comment 6:　Line 132-135: this data only questionnaires base please justify with diagnostic based results.

Our reply:

Thank you for your comment. Building-related symptoms (BRSs) and Sick building syndrome (SBS) are usually defined as the occurrence of nonspecific symptoms among populations in specific buildings. In epidemiological studies worldwide, BRSs and SBS are often diagnosed using questionnaires, such as the MM040 (Andersson, K., 1998). In this study, we diagnosed BRSs and SBSs using a questionnaire developed based on the “Consultation Manual for Trace Chemicals in the Indoor Environments” issued by the Japanese MHLW. We described the questionnaires in the “Questionnaire” section.

Comment 7: Line 199-201: In which parameters used for concluding results as health is good, explain clearly.

Our reply:

Thank you for the comment. Regarding the health of the day, we asked participants to judge their physical condition subjectively. We revised the “Questionnaire” section as follows:

“The participants were asked to subjectively score their health status on the day of the evaluation using a scale of 1–4: 1, very good; 2, good; 3, not good; or 4, bad. We defined 1 and 2 as “good” and 3 and 4 as “bad.””

Comment 8: 

Line 201: history of allergy- which types of reports you have for individuals motioned.

In table shown the data for personal factors, one of them physical condition most of them health and not any major health issues reported, until 50 % of them are have medical history of allergy. In pervious mentioned all individuals participate in this study are healthy explain it.

Our reply:

Thank you for the comment. In this study, the most common “medical history of allergy” was hay fever. Most symptoms of hay fever appear in early spring (February to May) in Japan. As this study was conducted from May to October, most participants with a history of hay fever had no symptoms at the time of the test. Thus, they may have evaluated themselves as healthy on those days. However, we changed “healthy volunteers” to “participants.”

Comment 9: 

Discussion section updated with recent and more related paper; this portion need improvement.

Our reply:

Thank you for the comment. We updated the Discussion with recent and related papers.

Comment 10: 

Rewrite abstract and conclusion section.

Our reply:

Thank you for the comment. We rewrote the abstract and conclusion.

Reviewer ＃２

Comment: 

The work described in the manuscript is based on questionnaire, air sampling and odor activity testing. Data are methodically and accurately analyzed. The conclusions are made basically reflecting to the results formulated.

Our reply:

We sincerely appreciate your review and comments.

We would like to take this opportunity to express our sincere thanks to the reviewers who identified areas of the manuscript that needed corrections or modifications. We would also like to thank you for allowing us to resubmit a revised copy of the manuscript. We hope that the revised manuscript is acceptable for publication in PLOS ONE.

Sincerely,

Kayo Tsumura 

Center for Preventive Medical Sciences, Chiba University, 1-33 Yayoi-tyou, Inage-ku, Chiba 263-8522, Japan 

Tel.: +81-43-290-3878 

Email Address: tsumu-kayo@chiba-u.jp

---

## [Editor Report · Decision Letter 1]

14 Dec 2022

Is indoor environment a risk factor for building-related symptoms?

PONE-D-22-23122R1

Dear Dr. Tsumura,

We’re pleased to inform you that your manuscript has been judged scientifically suitable for publication and will be formally accepted for publication once it meets all outstanding technical requirements.

Kind regards,

Rajeev Singh

Academic Editor

PLOS ONE
---

## [Editor Report · Acceptance letter]

17 Jan 2023

PONE-D-22-23122R1 

Is indoor environment a risk factor of building-related symptoms? 

Dear Dr. Mori:

I'm pleased to inform you that your manuscript has been deemed suitable for publication in PLOS ONE. Congratulations! Your manuscript is now with our production department. 

Kind regards, 

on behalf of

Dr. Rajeev Singh 

Academic Editor

PLOS ONE